# Variations in the Distribution and Genetic Relationships among *Luciola unmunsana* Populations in South Korea

Tae-Su Kim [1], Kwanik Kwon [2] and Gab-Sue Jang [1,*]

1 Department of Life Sciences, College of Natural Science, Yeungnam University, Gyeongsan 38541, Korea; taesunim@ynu.ac.kr
2 Research Center for Endangered Species, National Institute of Ecology, Yeongyang 36531, Korea; econlearn@nie.re.kr
* Correspondence: sunside@ynu.ac.kr

**Abstract:** The firefly species *Luciola unmunsana* was first discovered on the Unmunsan Mountain in Cheongdo-gun, Gyeongsangbuk-do, South Korea and consequently named after the mountain. The population and habitats of this once-abundant species have recently decreased significantly due to light and environmental pollution caused by industrialization and urbanization. This study investigated the distribution and density of *L. unmunsana* around the ecological landscape conservation area of the Unmunsan Mountain. Additionally, we conducted molecular experiments on regional variations, genetic diversity and phylogenetic relationships among the various populations of *L. unmunsana* in South Korea. The genetic relationships among populations were also analyzed using mitochondrial DNA by collecting 15 male adults from each of the 10 regions across South Korea selected for analysis. Differences were observed between populations in the east, west and south of the Baekdudaegan Mountain Range. The firefly populations collected from the eastern region, which included Gyeongsang-do, showed a close genetic relationship with fireflies collected from the Unmunsan Mountain. Thus, the findings of this study can be used as baseline data for re-introducing *L. unmunsana* to the Unmunsan Mountain.

**Keywords:** firefly; *Luciola unmunsana*; spatial distribution; mitochondrial DNA; genetic relationship



## 1. Introduction

Urban population growth and accompanying urban expansion have rapidly progressed in the Korean Peninsula since the end of the Korean War in 1953 [1]. This rapid urbanization has played a role in environmental pollution and habitat modification in South Korea [2]. Habitat loss and light pollution are known drivers of insect decline [3–5]. In particular, firefly populations and their habitats have considerably decreased due to artificial light [3–6], land-use changes [3], use of pesticides [4,7,8] and habitat modification [9] caused by rapid industrialization and urbanization [10]. Although fireflies were common insect species in South Korea several decades ago, their current numbers have become severely limited as a result of habitat modification by urbanization [11].

Only eight firefly species have been recorded in the Korean Peninsula [12], while they constitute more than 2000 species in 100 genera worldwide [13]. Three species are recognized in Korea in the genus *Luciola* of Luciolinae: *L. lateralis*, *L. unmunsana* and *L. papariensis* [14]. Among them, *L. unmunsana* Doi 1931, is a species endemic to Korea [15]. It was first discovered on Mt. Unmun in 1931 by the Japanese scholar Doi and has been named after the mountain [15]. Several genetic studies using luciferase genes showed that *L. papariensis* and *L. unmunsana* were the same species [16,17]. Phylogenetic studies of fireflies in Korea and Japan using luciferase and mitochondrial cytochrome oxidase I (COI) genes confirmed that the two species were synonymous [18–20]. *L. unmunsana* and *L. papariensis* are nearly identical morphologically [14,21]. Kim et al. [20] suggested that *L. papariensis* and *L. unmunsana* are not different species because the pronotal semicircular

speckle is polymorphic within species and sometimes differs between species. Kang [22] also pointed out that *L. papariensis* may be the same as *L. unmunsana*, or at least may not be distributed in South Korea based on topotypical specimens of *L. unmunsana* with the blackish semicircular speckle on the pronotum and collected from the type locality, Mt. Unmun [14].

According to two year-long research projects focused on *L. unmunsana* [23,24], it was revealed that the firefly has a one-year life cycle and lives as a larva for the most of its life, followed by the adult stage, which lasts only for approximately two weeks. The adults are normally active from late May to mid-July and their time of appearance varies depending on the altitude and climate of the habitat. Male and female adults show sexual dimorphism in shape. Males have two light emitters at the end of the abdominal segment, while females have only one. Furthermore, male adults can fly using their elytra and hind wings, but females cannot fly because their hind wings are functionally degenerated. This flightless morph in females impacts their ability to disperse [25].

Because of advances in sequencing and computational technologies, DNA sequences have become the major source of new information for advancing our understanding of evolutionary and genetic relationships [26]. Several types of DNA sequences have been employed for molecular phylogenetics and population genetics [27]. The mitochondrial cytochrome oxidase I gene (COI) has often been used to identify genetic variations within a species [28–31], phylogenic relationships [14,32,33] and the rate of evolution [34,35]. The mitochondrial cytochrome oxidase II gene (COII) has been used successfully to study population genetic structure and population history of a wide range of insect species [36–39]. The mitochondrial 16s rDNA sequence has been commonly used to clarify interspecific or intraspecific variations [28,40–42].

We hypothesized the occurrence of genetic variations among the *L. unmunsana* populations due to the flightless morphs in females, depending on the presence of geographical barriers between local populations (e.g., rivers, mountains). In this study, we investigated the spatial distribution of *L. unmunsana* populations and performed DNA sequencing using COII and 16s rDNA genes of *L. unmunsana* to identify the geographical variations among its populations.

## 2. Materials and Methods

### 2.1. Sampling Sites for L. unmunsana

The distribution of *L. unmunsana* populations were surveyed for three years (2012 to 2014) from May to July, including one year of preliminary survey in 2012 before confirming the exact study locations [23,24]. We chose 10 sampling regions: Jeju Island; Busan city; Gyeongsan-si, Mt. Unmun and Cheongdo-gun in Gyeongsangbuk-do; Goesan-gun and Okcheon-gun in Chungcheongbuk-do; Asan-si in Chungcheongnam-do; Jangseong-gun in Jeollanam-do; and Namyangju-si in Gyeonggi-do (Figure 1). All *L. unmunsana* individuals in these areas were confirmed to have an orange-red color pattern without a blackish speckle on the pronotum (Figure 2). In each region, a line census was conducted from 21:00 to 24:00 local time by two or more technicians with a mechanical counter (VA-1100HC) to assess the firefly population in the area. From this data, population averages were determined. Target areas were mapped using ArcMap ver. 10.1, as shown in in Figure 1. Soil organic matter and atmospheric humidity were measured to describe *L. unmunsana's* preferred habitat. Soil was sampled at four major sites (Mt. Unmun, Jeju Island, Pusan and Namyangju) which were well known for *L. unmunsana* habitat. The soil organic matter in each sample was determined using the process described by Heiri et al. [43]. We also used a hygrometer (DT-172, Sunshi Process Systems, Deccan gymkhana pune, India) to monitor atmospheric humidity at three major sites (Mt. Unmun, Igidae in Busan and Hannam-ri in Jeju).

## 2.2. DNA Extraction

Fifteen male adults were sampled for genetic analysis from each of the 10 regions (Table 1). The collected adults were first immersed in distilled water (DW) at the site for 2 h to remove ethanol. Subsequently, the samples were finely ground using a tissue homogenizer and BioMasher II Grinder (BIOFACT Inc., Daejeon, Korea) to completely homogenize the tissues. This process was performed using the HiGeneTM Genomic DNA Prep Kit for Animal Tissue (BIOFACT Inc., Daejeon, Korea). DNA concentrations from the samples were checked using electrophoresis (Takara Bio Inc., Shiga, Japan) on a 1.5% agarose gel.

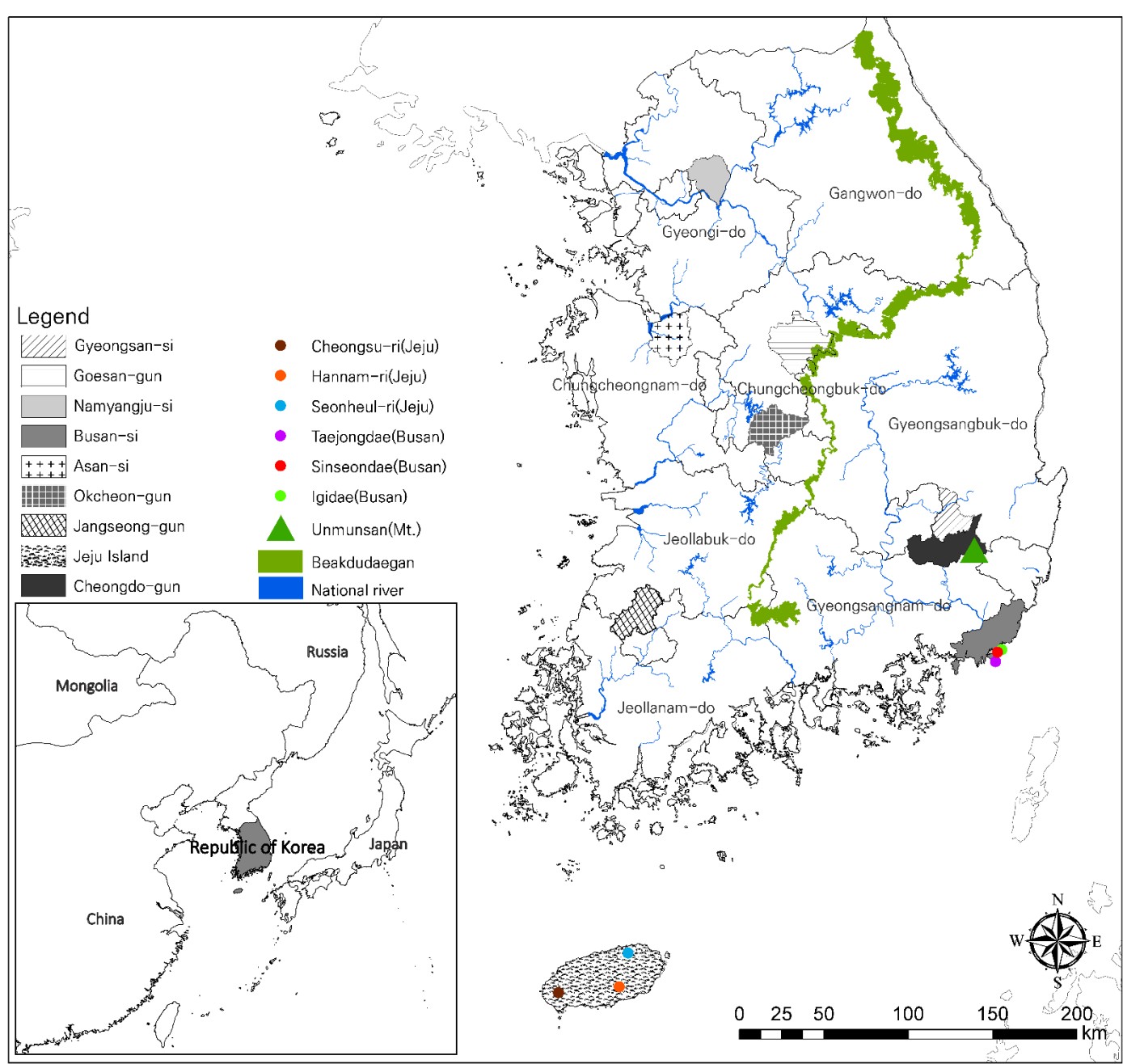

**Figure 1.** Study sites for *L. unmunsana*.

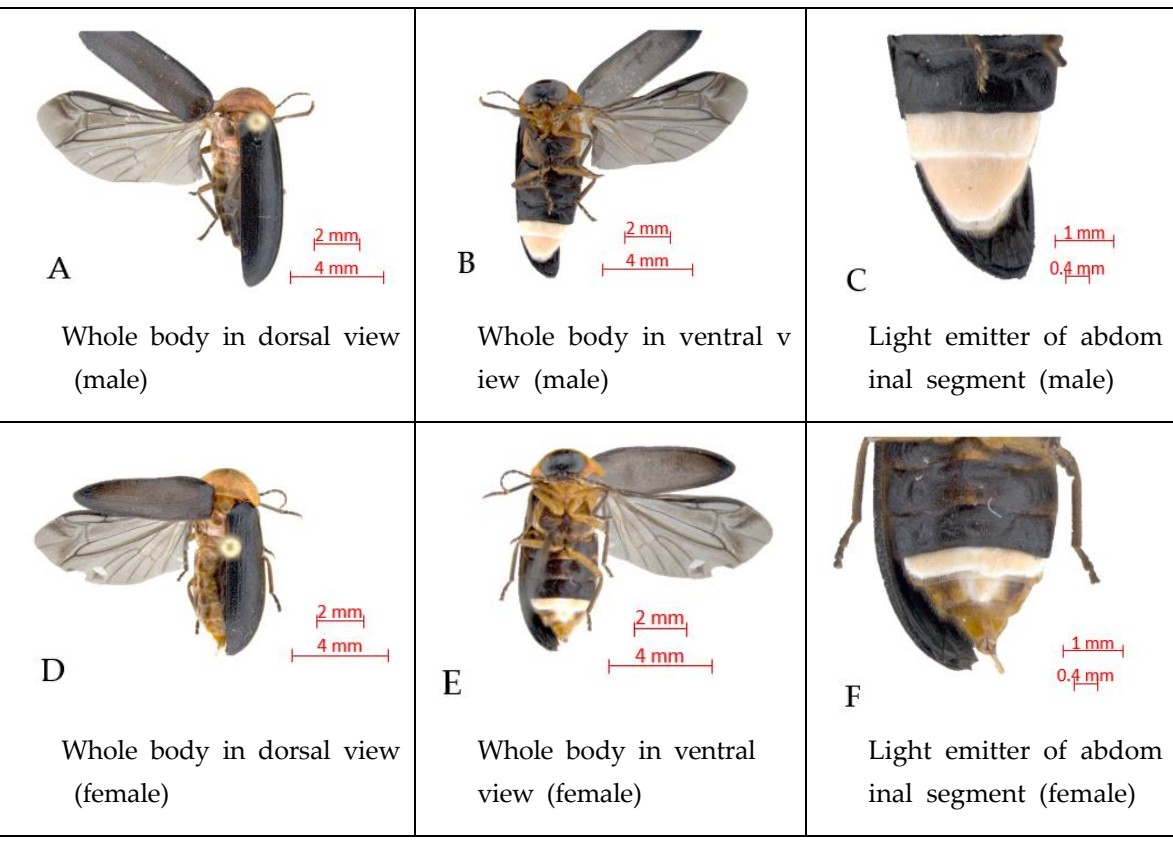

**Figure 2.** Views of *L. unmunsana* male and female adults.

**Table 1.** *L. unmunsana* sampling information used for genetic analysis.

| No. | Sampling Site (Population) | | | Sampling Date | Haplotype | | PCR Success | |
|---|---|---|---|---|---|---|---|---|
| | | | | | 16s rRNA (9) Hd: 0.9566 | COII (28) Hd: 0.9566 | 16s rRNA | COII |
| 1 | Asan-si | | 15 | 14.05.31 | H1 | H1, H2 H3. H4 | 15 | 15 |
| 2 | Cheongdo-gun | | 15 | 14.06.08 | H2 | H5, H6 H7, H8 | 15 | 14 |
| 3 | Busan city | Igidae | 5 | 13.06.04 14.06.05 | H2 | H9 H10 | 15 | 12 |
| | | Sinseondae | 5 | 13.06.19 14.06.05 | | | | |
| | | Taejongdae | 5 | 13.06.19 14.06.05 | | | | |
| 4 | Gyeongsan-si | | 15 | 14.05.30 | H2 | H19, H20 H21, H22 | 15 | 14 |
| 5 | Mt. Unmun | | 15 | 13.06.10 14.05.29 14.07.14 14.07.16 14.07.18 | H2 H9 | H28 | 15 | 13 |
| 6 | Namyangju-si | | 15 | 13.06.19 | H8 | H24, H25 H26, H27 | 15 | 15 |

**Table 1.** *Cont.*

| No. | Sampling Site (Population) | | | Sampling Date | Haplotype | | PCR Success | |
|---|---|---|---|---|---|---|---|---|
| | | | | | 16s rRNA (9) Hd: 0.9566 | COII (28) Hd: 0.9566 | 16s rRNA | COII |
| 7 | Okcheon-gun | | 15 | 13.06.08 14.05.31 | H7 | H23 | 15 | 15 |
| 8 | Goesan-gun | | 15 | 14.05.31 | H1 H6 | H17 H18 | 15 | 15 |
| 9 | Jeju Island | Cheongsu | 5 | 13.07.08 14.06.26 | H3 H4 | H11 H12 H13 | 15 | 15 |
| | | Hannam | 5 | 13.06.24 14.06.27 | | | | |
| | | Seonheul | 5 | 13.06.26 14.06.29 | | | | |
| 10 | Jangseong-gun | | 15 | 14.06.05 | H5 | H14, H15 H16 | 15 | 15 |

*2.3. Sequencing*

The extracted DNA was diluted 10 times for sequence amplification using the polymerase chain reaction (PCR) process. Subsequently, 1.0 μm DNA template was amplified using a 10× PCR buffer, 2.5 mM dNTP, 5 U Taq (DNA Polymerase), Band Doctor solution, 10 pmole primer set (TK2-J-3037, TK-N-3785/LR-J-12887-1, LR-N-13398-1) and rinsed thrice with DW. The primers were selected based on the studies of Suzuki and Kim [18,44,45]. Furthermore, some sequences were modified for 16s rRNA analysis (Table 2).

**Table 2.** Primers used in the analysis of genetic relationships.

| Gene | Sequence | Reference |
|---|---|---|
| TK2-J-3037 TK-N-3785 | COII 5′-ATGGCAGATTAGTGCAATGG-3′ 5′-GTTTAAGAGACCAGTACTTG-3′ | [44,45] |
| LR-J-12887-1 LR-N-13398-1 | 16s rRNA 5′-CCGGTTTAAACTCAGATCATGT-3′ 5′-TGCCTGTTTATTAAAAACAT-3′ | [18] |

The conditions of the PCR process were as follows: pre-denaturation at 94 °C for 5 min, denaturation at 94 °C for 1 min, annealing at 60 °C for 1 min and extension at 72 °C for 1 min. These steps were repeated 30 times. In addition, a final extension at 72 °C for 7 min was followed by cooling at 4 °C for 10 min. The amplified PCR products were loaded on a 1.5% agarose gel, mixed with an eco-dye and checked under UV light. Samples which exhibited faint or invisible bands during electrophoresis were subjected to PCR again and the entire process was repeated. PCR products with normal bands were purified using a HiGeneTM PCR Purification Kit and sequenced along with the primer (Table 2).

Sequences of cytochrome oxidase II (COII) and 16s rRNA genes were sorted and edited using Geneious ver. 5.6.7 and MEGA X. The sequences were identified as *L. unmunsana* through a basic local alignment search tool (BLAST) by the National Center for Biotechnology Information (NCBI). Haplotype and haplotype diversity were calculated using DNA sp ver. 5. Different sequences were considered as different haplotypes. Accordingly, a network of sequence haplotypes was prepared using TCS ver. 1.21. Additionally, a maximum likelihood (ML) method was created in MEGA X based on the gamma distributed with an invariant sites (G + I) pattern by the general time reversible (GTR) model [46]. To determine the most appropriate base substitution model for the Bayesian inference (BI)

method, a model test was performed using Jmodeltest 2.1.7. Then, based on the highest
score using the Akaike information criterion (AIC), the HKY + g and the general time
reversible models were selected for COII and 16s rRNA analysis, respectively. Based on
these models, Bayesian analysis was performed using MrBayes 3.2.3 [47,48] until the value
after 3,000,000 generations was less than 0.05. The tree sampling option of each model was
analyzed using the HKY + g and GTR models and the log-likelihood value of each model
was analyzed using the Markov Chain Monte Carlo (MCMC) method. A phylogenetic tree
was generated from the results using FigTree ver. 1.4.4.

### 2.4. AMOVA Test

The genetic diversity of each population was analyzed through an analysis of molec-
ular variance (AMOVA) test and pairwise differences were determined using Arlequin
ver. 3.5.1 [48,49]. Among the 10 selected study sites, the west and east sides of the Baekdu-
daegan Mountain Range (Figure 1) and Jeju Island were identified to have three separate
populations. This was based upon the assumption that the female *L. unmunsana's* inability
to fly could have separated the populations due to mobility limitations. The genetic vari-
ations among groups, among populations within a group and within populations were
determined using the standard AMOVA test. Finally, genetic distances between haplotypes
were assessed based on pairwise differences.

## 3. Results and Discussion

### 3.1. Population Density of L. unmunsana

Surveys during two consecutive years (2013 and 2014) showed that several villages in the
Jeju island (e.g., Cheongsu and Hannam) in South Korea had relatively higher *L. unmunsana*
populations. Cheongsu had the largest population counts (452 in 2013, 327 in 2014) during
100-m line censuses. Outside of the Jeju island, Okcheon had the largest *L. unmunsana* inland
population. There, 144 individuals were counted during the 100-m line census of 2013.
Though Mt. Unmun was famous as type locality for *L. unmunsana*, populations there were
lower than at other locations (7 to 12 individuals in 2013 and 4 to 28 individuals in 2014).
Gyeongsan and Jangseong had slightly higher populations in a one-year, 100-m census with
over 70 individuals counted at both areas (Table 3 and Figure 3).

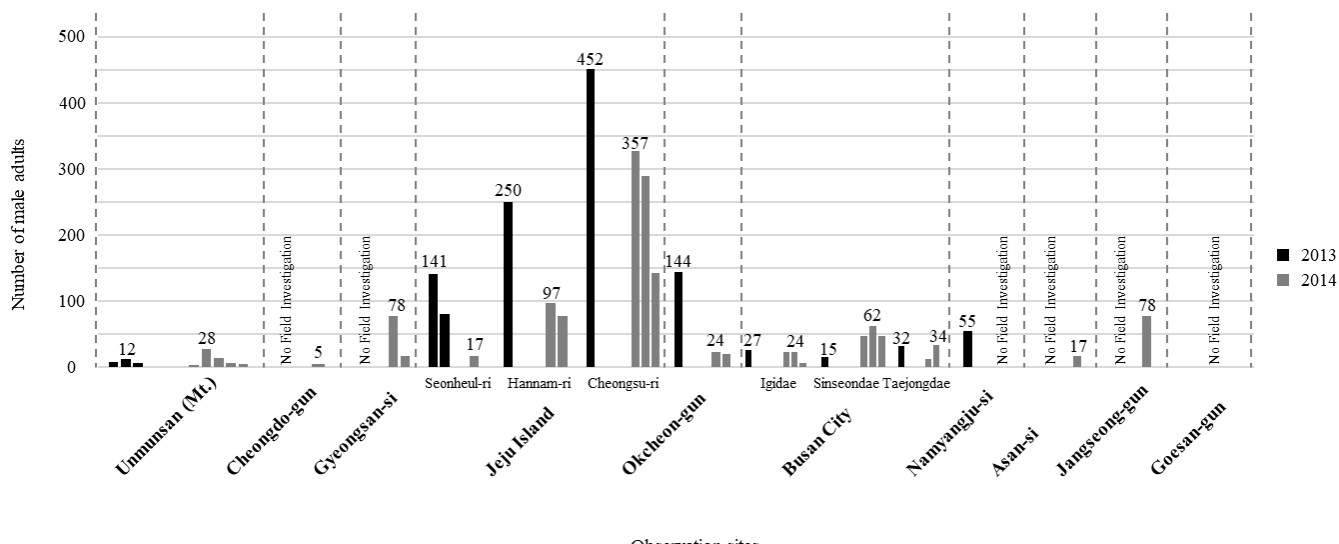

**Figure 3.** Population density of *L. unmunsana* in South Korea.

**Table 3.** Census results of *L. unmunsana* in South Korea.

| No. | Observation Sites | Date (2013) | No. of Males | Date (2014) | No. of Males |
|---|---|---|---|---|---|
| 1 | Mt. Unmun | 6 June 2013 | 8 | 8 June 2014 | 4 |
|  |  |  |  | 29 June 2014 | 28 |
|  |  | 14 June 2013 | 12 | 14 July 2014 | 14 |
|  |  |  |  | 16 July 2014 | 7 |
|  |  | 18 June 2013 | 7 | 18 July 2014 | 5 |
| 2 | Cheongdo-gun | No field investigation | | 8 June 2014 | 43 |
| 3 | Gyeongsan-si | No field investigation | | 30 May 2014 | 78 |
|  |  |  |  | 7 June 2014 | 17 |
| 3 | Seonheul-ri (Jeju) | 26 June 2013 | 141 | 29 June 2014 | 17 |
|  |  | 29 June 2013 | 81 |  |  |
| 4 | Hannam-ri (Jeju) | 27 June 2013 | 250 | 27 June 2014 | 97 |
|  |  |  |  | 1 July 2014 | 78 |
| 5 | Cheongsu-ri (Jeju) | 34 June 2013 | 452 | 26 June 2014 | 327 |
|  |  |  |  | 28 June 2014 | 289 |
|  |  |  |  | 30 June 2014 | 143 |
| 6 | Okcheon-gun | 7 June 2013 | 144 | 31 May 2014 | 24 |
|  |  |  |  | 7 June 2014 | 20 |
| 8 | Igidae (Busan) | 5 June 2013 | 27 | 5 June 2014 | 23 |
|  |  |  |  | 10 June 2014 | 24 |
|  |  |  |  | 11 June 2014 | 7 |
| 9 | Sinseondae (Busan) | 6 June 2013 | 15 | 5 June 2014 | 47 |
|  |  |  |  | 10 June 2014 | 62 |
|  |  |  |  | 11 June 2014 | 47 |
| 10 | Taejongdae (Busan) | 4 June 2013 | 32 | 5 June 2014 | 12 |
|  |  |  |  | 11 June 2014 | 34 |
| 11 | Namyangju-si | 19 June 2013 | 55 | No field investigation | |
| 12 | Asan-si | No field investigation | | 31 May 2014 | 17 |
| 13 | Jangseong-gun | No field investigation | | 5 June 2014 | 78 |
| 14 | Goesan-gun | No field investigation | | | |

We found that every site having a relatively large *L. unmunsana* population (e.g., three areas on Jeju Island) had more organic matter in the soil. Results of soil organic matter sample analyses from several sites showed that three regions in Jeju had higher soil organic matter (Table 4) than other regions (including Unmun Mt., Sinseondae and Namyangju). This suggests that *L. unmunsana* prefers soils abundant in organic matter. Furthermore, they appear to prefer topsoil with a pronounced organic litter layer.

The field survey also revealed that preferred habitats of *L. unmunsana* included forest edges with abundant water resources such as a lake or continuously wet valley (Figure 4). This suggests that humidity, as well as soil organic matter, may be a key factor in habitat preference. However, survival of *L. unmunsana* populations in regions with steep and/or rough topography (e.g., Mt. Unmun, three regions in Busan, Gyeongsan, etc.) suggests that *L. unmunsana* can survive regardless of topsoil and humidity conditions. The population occurrence of *L. unmunsana* is along the Mt. Unmun's biggest watershed where there is a

continuously wet valley that contributes humidity for its habitat, even without significant soil organic matter (Figure 4h). Three regions in Busan (Igidae, Sinseondae and Taejongdae) also supported small populations of *L. unmunsana* despite having less than ideal soil and topographic conditions. These regions receive moist air carried by sea winds from the East Sea (Figure 4c).

Though there is no major river or lake on Jeju Island, the humidity derived from plentiful surface water running through a wide buffer strip along Mt. Halla (Figure 4g) reliably substitutes for the humidity otherwise associated with a continuously wet valley (Figure 4a–f,h). As Figure 5 shows, the three major habitats for *L. unmunsana* (Okcheon and Mt. Unmun for inland conditions and Jeju for island conditions) had relatively high daily mean humidity levels.

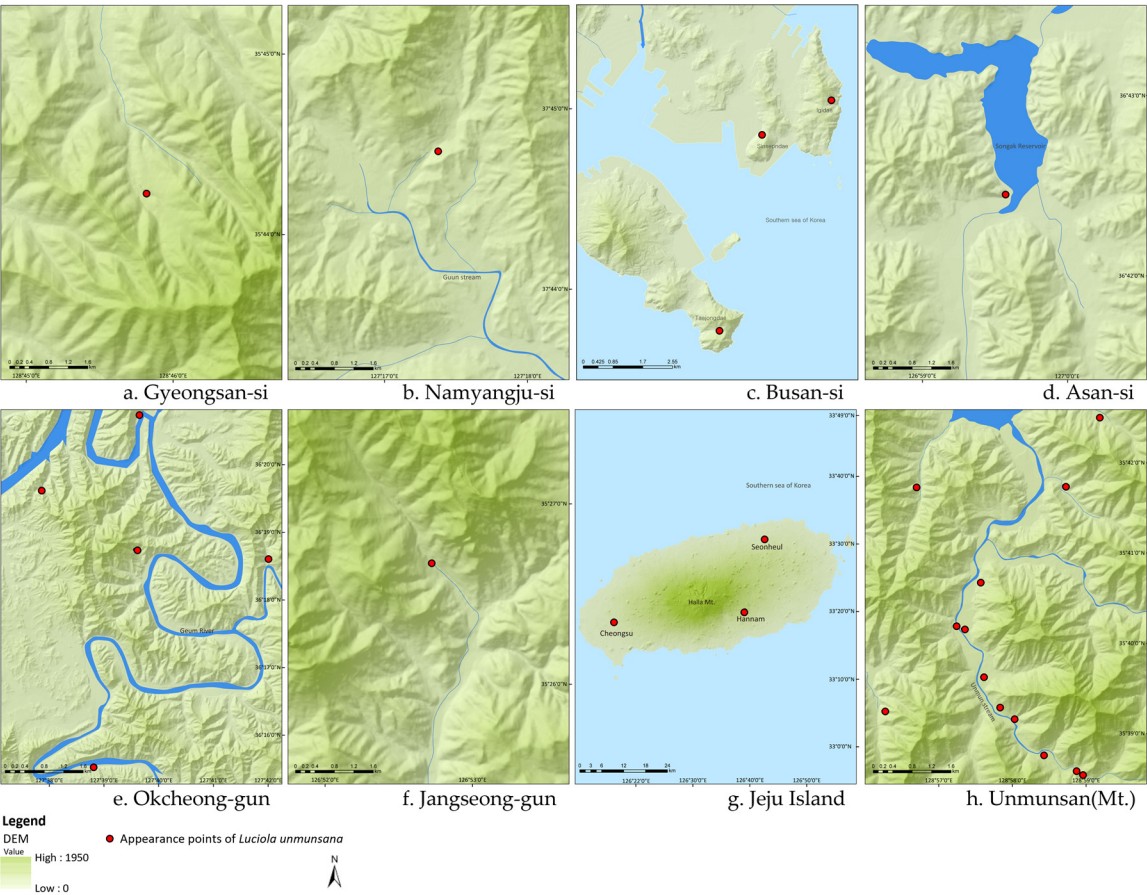

**Figure 4.** *L. unmunsana* habitats with continuously wet valleys (except for Jeju Island).

**Table 4.** Comparing organic matter and color of soil in the representative habitats of *L. unmunsana*.

| Site Name | DW | WLI | LOI (% wt. loss) | TOC | OM (%) | Soil Color | |
|-----------|----|----|------------------|-----|--------|-----------|---|
| Cheongsu | 10 | 3.8 | 38 | 17.00 | 29.31 | | |
| Hannam | 10 | 5.1 | 51 | 22.96 | 39.58 | Brownish black | |
| Seonheul | 10 | 4.4 | 44 | 19.75 | 34.05 | | |
| Mt. Unmun | 10 | 0.4 | 4 | 1.43 | 2.46 | Grayish yellow | |
| Igidae | 10 | 1.5 | 15 | 6.47 | 11.15 | Olive brown | |
| Sinseondae | 10 | 0.8 | 8 | 3.26 | 5.627 | Yellowish brown | |
| Namyangju | 10 | 0.3 | 3 | 0.97 | 1.679 | Dark grayish yellow | |

DW: dry weight; WLI: Wt. loss on ignition; LOI (%): loss on ignition; TOC: total organic carbon; OM: organic matter.

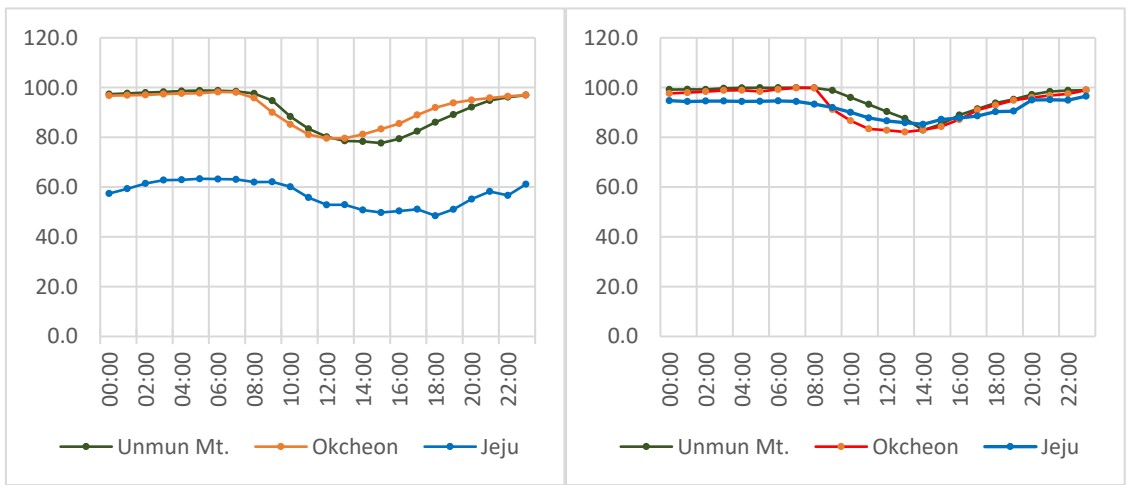

**Figure 5.** Daily mean atmospheric humidity in June (**left**) and July (**right**), 2014.

### 3.2. Genetic Analysis

PCR and sequencing were performed on the mitochondrial COII and 16s rRNA genes of 150 *L. unmunsana* fireflies. Of these, results from 143 and 150 fireflies, respectively, were viable. The species were identified as *L. unmunsana* through the BLAST search in the NCBI database. Furthermore, COII had a length of 615 bp and 27 haplotypes (H1–H27), while 16s rRNA exhibited a length of 468 bp and 9 haplotypes (H1–H9). A network was developed using TCS 1.21 software. We conducted a systematic analysis using Bayesian inference (BI) methods and maximum likelihood (ML); with *L. lateralis*, *L. cruciata*, *Hotaria parvula* and *Hotaria thushimana* as outgroups. A network developed using the TCS program with the 27 haplotypes of the COII gene indicated a clear distinction between the populations of Gyeongsang-do, Chungcheong-Gyeonggi, Jeollabuk-do, Jeollanam-do and Jeju Island, separated by the Baekdudaegan Mountain Range (Figure 6). A characteristic of the TCS

program is that the populations are not interconnected after the 95% connection limit is exceeded. Therefore, the disconnection between the Jeju Island, Jeollabuk-do, Jeollanam-do, Chungcheong-Gyeonggi and Gyeongsang-do regions in the network indicates that the populations are distinct from one another. Furthermore, small populations around each constellation-shaped population indicated that the firefly populations stabilized after past mutations that occurred within the group. By contrast, although the groups formed around Gyeongsang-do, Chungcheong-do, Gyeonggi-do, Jeollabuk-do, Jeollanam-do and Jeju Island in the network based on the 16s rRNA gene analysis were similar to those observed based on the COII results, no significant differences in the distances between the groups were observed, despite regional distances (Figure 7). The possible reasons for this are as follows: First, due to the differences in the number of haplotypes in the gene, the short 16s rRNA sequence showed a limited number of variations. Second, the frequency of haplotypes shared among the groups in each region was low. The small number of haplotypes shared among the groups, combined with the unique haplotypes in each region, shows that the degree of differentiation in each region had progressed considerably.

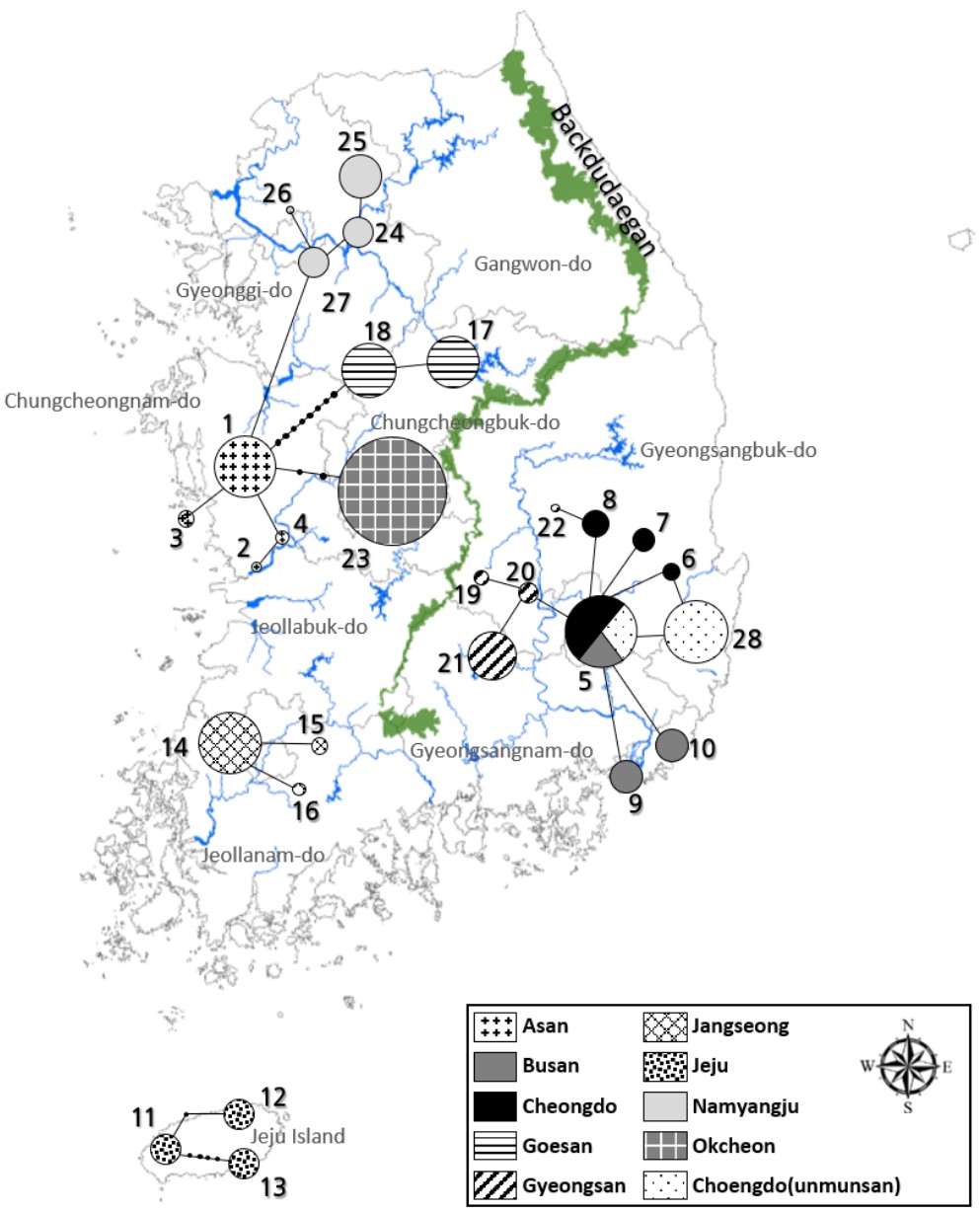

**Figure 6.** Haplotype network based on the mitochondrial DNA COII sequence of four local *L. unmunsana* groups.

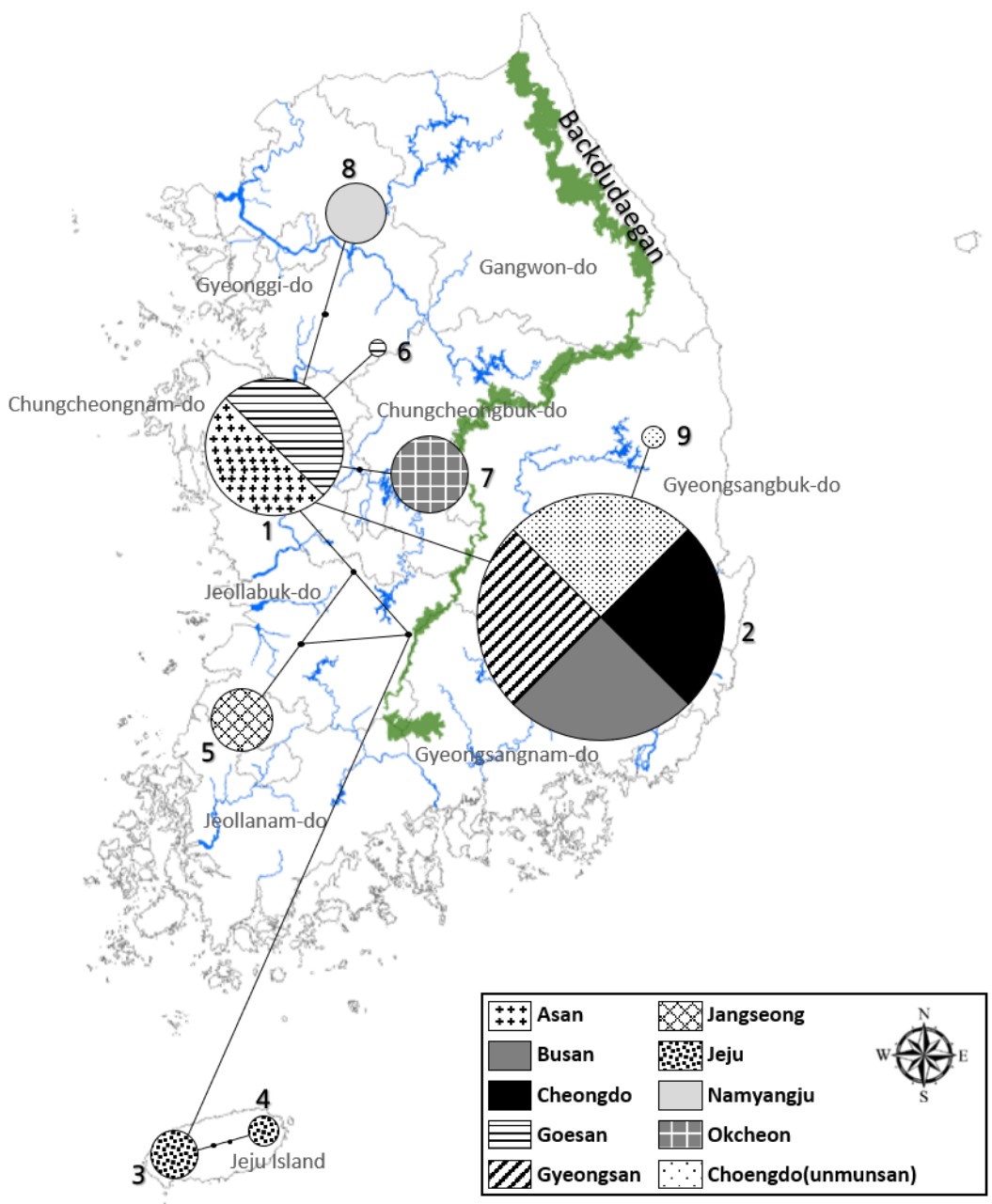

**Figure 7.** Haplotype network based on the sequence mtDNA 16s rRNA of *L. unmunsana*.

The systematic analysis of the haplotypes of the COII and 16s rRNA genes resulted in one group, comprising firefly populations of the Mt. Unmun, Cheongdo-gun, Busan City and Gyeongsan-si in the BI and ML trees, for both COII and 16s rRNA genes, similar to the network results (Figures 8–10). Furthermore, every node value in the BI and ML analyses was supported by 70% or higher on the BI and ML trees using the COII and 16s rRNA genes. However, the correlations between the regions in the trees were not distinct because of low genetic variations. A distinct genetic distance was observed in the network generated from haplotype-based analysis of the Jeju Island populations. However, in the ML analyses, the island populations appeared closer to those in the eastern region of Baekdudaegan, which is unlikely. This is because few or no haplotypes were shared by the populations and each population had already undergone genetic variations over previous years, resulting in unique regional haplotypes.

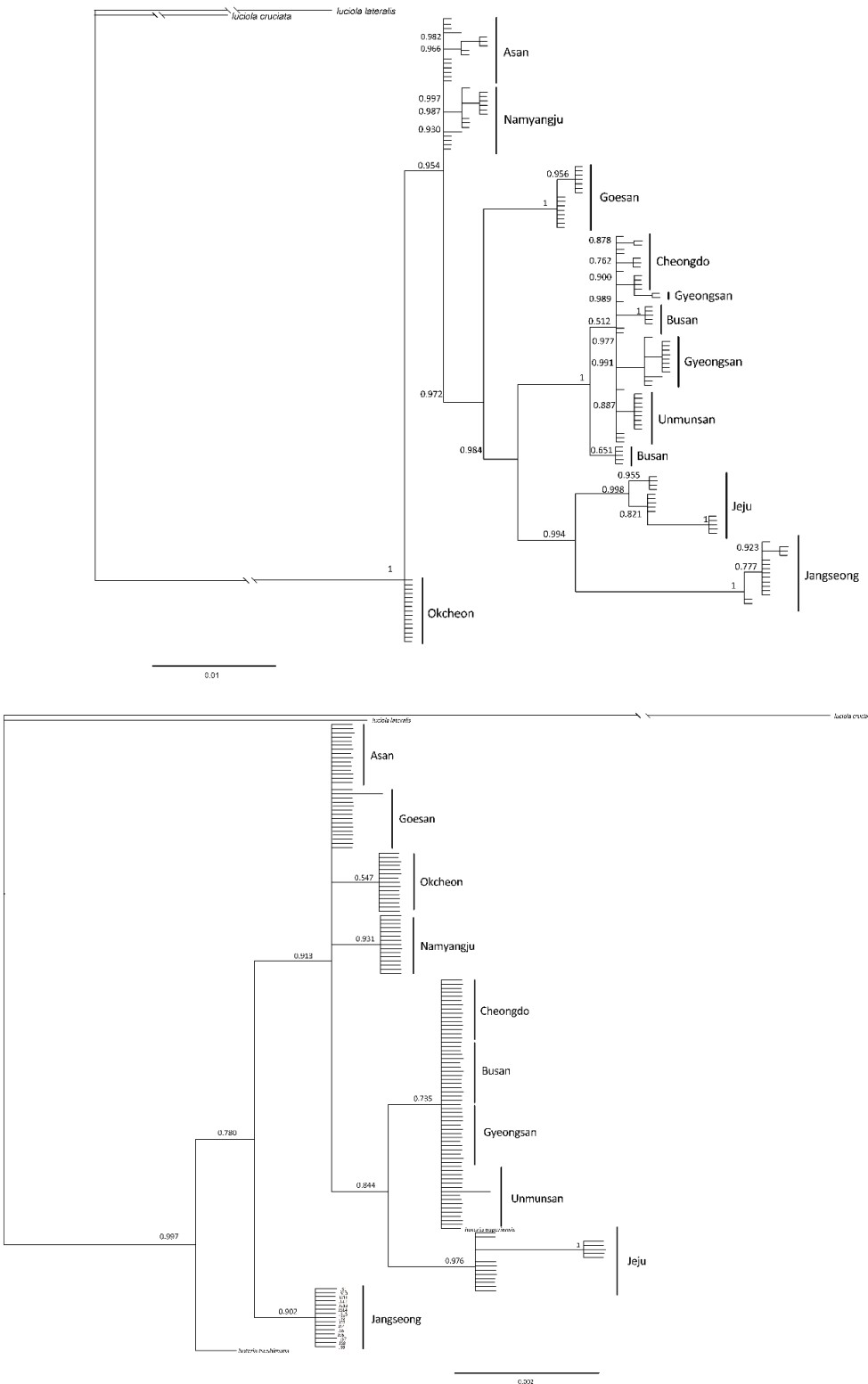

**Figure 8.** Phylogenetic analysis of 27 mitochondrial COII gene (**upper tree**) and 9 mitochondrial 16s rRNA gene (**under tree**) of *L. unmunsana*. The trees were developed by the Bayesian inference method using the Mrbayes software.

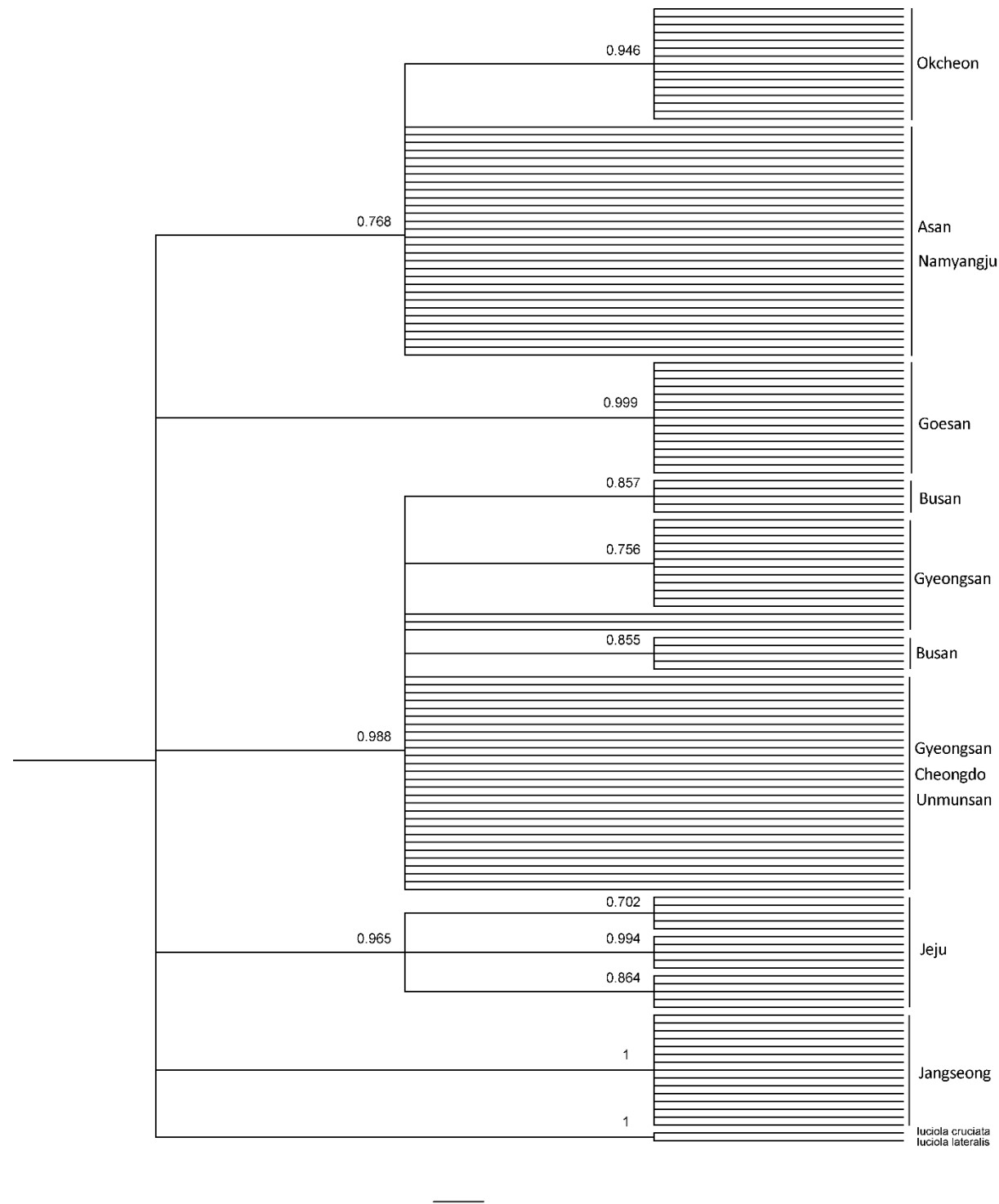

**Figure 9.** Phylogenetic analysis of 27 mitochondrial COII gene of *L. unmunsana*. The tree was developed by the ML method using the MEGA software.

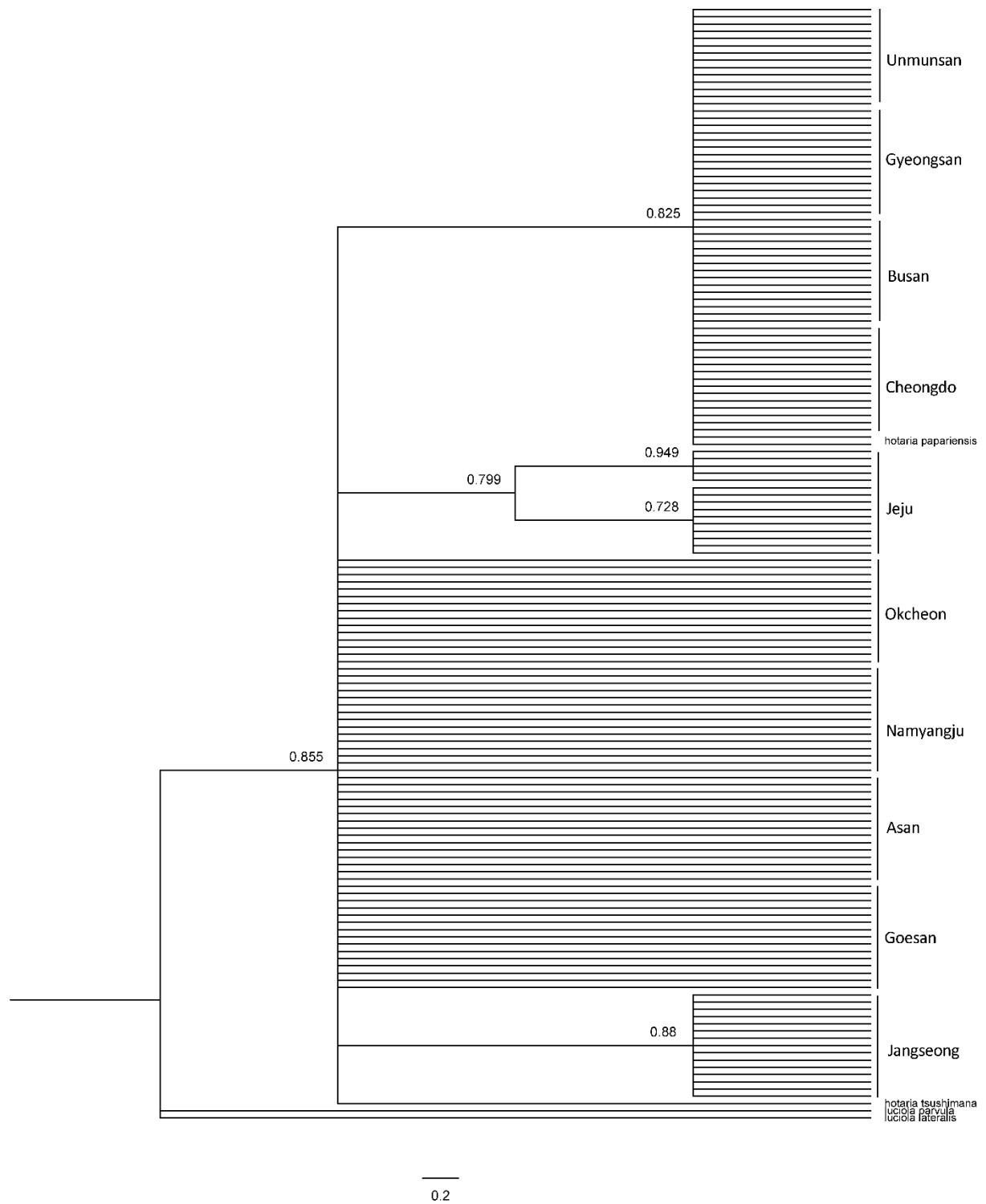

**Figure 10.** Phylogenetic analysis of 9 mitochondrial 16s rRNA gene of *L. unmunsana*. The tree was developed by the ML method using the MEGA software.

*3.3. AMOVA Test*

Groups were artificially set up in all the regions based on the network and BI and ML tree methods were used to analyze the genetic variations through the AMOVA test. However, the network results showed that the COII gene was becoming isolated. Hence, four groups were established around the Baekdudaegan Mountain Range: the eastern region (Mt. Unmun, Cheongdo-gun, Gyeongsan-si and Busan City); western region

(Goesan-gun, Asan-si, Okcheon-gun and Namyangju-si); southern region (Jangseong-gun); and the Jeju Island, which was established separately. This shows that isolation must have progressed further on the island than in the inland areas. Consequently, the percentage of variation was 69.54% among the groups, 23.9% among populations within groups and 6.47% within populations (Table 5). This suggests that genetic variations between the groups have progressed to some degree, whereas the degree of variation among populations in each group was low. The genetic distance among the groups implied that differentiation was underway as the species had stabilized much earlier. This was similar to the initial assumption of mobility being limited by flightless female adults. The relatively low frequency of genetic variation among populations in each group indicated that the gene transfer rate was low. Pairwise difference analysis showed values close to 0 for Mt. Unmun, Busan City, Cheongdo-gun and Gyeongsan-si. This indicates close genetic distances, whereas the values for the other regions were close to 1, indicating wide genetic distances (Table 6). Because each group had unique haplotypes and few or no haplotypes were shared among the groups, the results of genetic diversity comparisons were marginally significant. Although the genetic distances or the values comparing genetic relationships among the regions were uncertain, the populations genetically closest to those found in the Mt. Unmun ecological landscape conservation area were identified.

**Table 5.** Hierarchical analysis based on the mitochondrial COII gene.

| Source of Variation | df | Sum of Square | VC | PV | FI | ST |
|---|---|---|---|---|---|---|
| Among group | 3 | 698.70 | 6.33 | 69.54 | 0.75444 | 0.00000 |
| Among populations within group | 6 | 188.31 | 2.18 | 23.99 | 0.92869 | 0.00000 |
| Within populations | 133 | 78.31 | 0.59 | 6.47 | 0.70960 | 0.00098 |
| Total | 142 | 965.32 | 91 | 100 | | |

df = degrees of freedom; VC = variation components; PV = percentage of variation; FI = fixation indices; ST = significance tests.

**Table 6.** Pairwise differences in the genes among the regional population by the distance method using the mitochondrial COII gene.

| Site | CD | UM | NC | BS | SD | AS | OC | GS | JS | JJ |
|---|---|---|---|---|---|---|---|---|---|---|
| CD | - | | | | | | | | | |
| UM | 0.3777 | - | | | | | | | | |
| NC | 0.5644 | 0.6667 | - | | | | | | | |
| BS | 0.3180 | 0.4609 | 0.5561 | - | | | | | | |
| SD | 0.9159 | 0.9392 | 0.8954 | 0.8807 | - | | | | | |
| AS | 0.9212 | 0.9455 | 0.9008 | 0.8868 | 0.6202 | - | | | | |
| OC | 0.9662 | 0.9864 | 0.9473 | 0.9390 | 0.8905 | 0.8677 | - | | | |
| GS | 0.9496 | 0.9691 | 0.9324 | 0.9220 | 0.9286 | 0.9262 | 0.9785 | - | | |
| JS | 0.9681 | 0.9794 | 0.9562 | 0.9503 | 0.9660 | 0.9699 | 0.9887 | 0.9800 | - | |
| JJ | 0.8717 | 0.8871 | 0.8575 | 0.8398 | 0.8601 | 0.8655 | 0.8980 | 0.8849 | 0.9080 | - |

CD = Cheongdo-gun; UM = Unmunsan Mt.; NC = Gyeongsan-si; BS = Busan City; SD = Namyangju-si; AS = Asan-si; OC = Okcheon-gun; GS = Goesan-gun; JS = Jangseong-gun; JJ = Jeju Island.

## 4. Conclusions

This study investigated the distributions of *L. unmunsana* male adults in the major regions of the Korean Peninsula, conducted genetic analysis of the male adults observed in each region and derived genetic relationships among the regional populations.

Among the regions in Korea where *L. unmunsana* population was detected, the Jeju Island had the highest *L. unmunsana* population counts, with densities of more than 300 male adults per unit area. Additionally, *L. unmunsana* appeared in other regions (including Busan City, Okcheon-gun, Gyeongsan-si, Namyangju-si, Jangseong-gun and Asan-si), but their density was approximately 30 to 50 male adults per unit area. *L. unmunsana* populations were nationally distributed and appeared from late May to mid-July. The time of

appearance varied regionally; from late May to mid-July in inland areas and from mid-June to mid-July on the Jeju Island. In particular, the fireflies appeared later on the Jeju Island, although it experiences a warm and humid climate in all seasons. Furthermore, results of genetic analyses indicated a difference in the genetic distance between populations on the Jeju Island and those in inland areas. However, these findings require further investigation.

Genetic analysis of male adults was performed for the 10 regions in South Korea using mitochondrial DNA to analyze the genetic relationships among the *L. unmunsana* populations. A genetic distance was observed between the east and west sides of the Baekdudaegan Mountain Range. A different genetic distance was observed in the Jeju Island. Because the habitats of fireflies are gradually decreasing, many local governments are attempting to preserve and restore firefly species in South Korea. Thus, establishing clear directions for restoration and introduction of fireflies, to maintain genetic diversity, is important from an ecological perspective.

**Author Contributions:** Conceptualization, G.-S.J.; methodology, T.-S.K. and K.K.; software, T.-S.K. and K.K.; validation, T.-S.K. and G.-S.J.; formal analysis, G.-S.J.; investigation, G.-S.J.; resources, T.-S.K. and K.K.; data curation, T.-S.K. and K.K.; writing—original draft preparation, T.-S.K., K.K. and G.-S.J.; writing—review and editing, T.-S.K., K.K. and G.-S.J.; visualization, T.-S.K. and K.K.; supervision, G.-S.J.; project administration, G.-S.J.; funding acquisition, G.-S.J. All authors have read and agreed to the published version of the manuscript.

**Funding:** This research was supported by Yeungnam University Research Grants in 2019.

**Institutional Review Board Statement:** Not applicable.

**Informed Consent Statement:** Not applicable.

**Data Availability Statement:** All of the sample data were gathered in fields and treated by the authors for this article.

**Conflicts of Interest:** The authors declare no conflict of interest.

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
