# Peer review of "Variations in the Distribution and Genetic Relationships among Luciola unmunsana Populations in South Korea"

_land, doi:10.3390/land10070730_

Round 1

Reviewer 1 Report

The authors present two separate but related findings in their paper.  The first, was the distribution and prevalence/density of male H. unmensana across 10 well-defined regions of South Korea over a two-year period. The second was the molecular analysis of 15 individual fireflies from each of these regions to determine the genetic relationship among these populations.  The results of the genetic distance data were interpreted with respect to how and when these populations were established.

This paper could be improved by more clearly targeting this paper to a specific audience. The genetic analyses and results are the more robust element of the study. The population density data will be of significance if used as a baseline for future surveys to monitor population changes. But these population results are too preliminary to provide any detailed insights into the environmental pressures that might be causing declines in firefly populations, or to suggest potential interventions.

Specific comments:

  • Three typographical errors should be corrected - "exhbied" line 179 should be "exhibited"; "suvey" line 215 should be "surveyed"; Line 215 refers to "Figure 4" which does not exist.
  • (Lines 33-38) The authors introduce begin their introduction by describing factors that potentially impact firefly populations. This should be better documented with citations. In addition, it might be valuable to highlight factors that might specifically be impacting the populations under study in this paper and to cite any relevant studies.
  • (Lines 93-94) What was the collection/storage protocol? The authors indicate that they used DW to wash out the ethanol, but do not indicate the concentration of ethanol used for storage after collection.
  • (Lines 170-192) The report of the two years of population survey data was very interesting. But it would be difficult to predict any pattern of population change from only these two years. Several factors might be effecting population densities and their measurements such as weather trends, ambient lighting (e.g., moonlight), sampling period. More extensive monitoring of populations would be needed to establish clear trends.  Historical anecdotal information may also supplement this, although it may be difficult to compare qualitative reports with actual density measurements.
  • The authors also speculate (187-189) that tourist load may be a factor impacting population reduction but no data is provided to support this.

Author Response

Dear author:

Thank you so much for your kind comments on our manuscript. Your comments were definitely helpful for us to modify our one. We have changed many parts depending on your comments and you will meet with a newer upgraded one.

Once again we appreciate your contribution to our manuscript.

  1. We tried to robust 1st part of the manuscript: spatial distribution of L. unmunsana in Korea with additional experiment data.
  2. (Lines 33-38) We modified this part and add additional references for this mention.
  3. (Lines 93-94) We used 95% alcohol to keep the specimen for genetic analysis. We mentioned it at the same part.
  4. (Lines 170-192) We added the condition of soil and soil organic matter to regulate humidity content in soil. 
  5. (187-189) We removed these lines. We also understand what you mentioned.

Reviewer 2 Report

Reviewer response to

Kim Kwong and Jang:

Variations in the Distribution and Genetic Relationships among Hotaria unmunsana populations in the Unmunsana Mountain South Korea.

General comments (detailed comments follow below)

  1. I like the basis for your investigation it is original (for S Korea), and well written and presented. However there are references you could use to enhance your approach as Kato et al. (2020) did a similar type of investigation with Luciola cruciata in Japan with the express desire to manage the firefly transplantation in nature conservation and regeneration, as you are doing. You do not follow taxonomic references which address the nomenclature.
  2. I guess that your basic knowledge of entomology and fireflies especially is a bit poor but happy to be proved wrong.
  3. You will need to decide why you are referring to this species as Hotaria and not Luciola. McDermott 1966 reduced Hotaria to a subgenus of Luciola. Kawashima et al. (2003) submerged Hotaria under Luciola for two Japanese species, and Jusoh et al. (2021) considered at best it could be regarded as a subgenus of However you will find that Korean researchers like Sim & Kwon 2000, Park et al. 2003 and Choi et al. 2002 all refer to it as Hotaria but probably just followed what others in their field has done. To list some more recent references from your own area Kim Park & Kim 2020 called it Luciola (they were however more concerned with how to manage species of Luciola and what to call them). There are other references you should access. If you want to regard it as a subgenus (which you would need to substantiate) then call it Luciola (Hotaria) unmunsana.
  4. Scientific names are always italicized, the species name begins with a lower case letter, and the first time they are mentioned you should give the name of the author. If you omit the date then you will not need to list that publication. You have the name incorrectly spelled in the title.
  5. Luciola lateralis is now Aquatica lateralis.
  6. How did you know you had unmunsana when you surveyed? How did you distinguish this from Luciola papariensis which also occurs in Korea?? You should mention this. Kang 2012 mentioned the colour pattern of the pronotum. So did Kim Park and Kim 2020 who said unmunsana and papariensis were nearly identical (they both have the flightless female) apart from the pattern on the pronotum. Or did you use light patterns?
  7. Your method of collecting identifying and counting has me confused. Perhaps you can indicate a bit more clearly. How did you know which firefly you counted and how to know if you counted them twice? Was this carried out at night? I am sort of assuming so. How did you identify them in the dark?

Detailed comments

Line 3 do not capitalize the first letter of unmunsana.

Line 24. Consequently is probably incorrect here. Use ‘as a result’ or just begin with ‘differences were observed”.

Line 27. Thus. Isn’t it hopefully they can be used? I would consider another word to thus.

Lines 33-39. You need a reference here and there are many that address this problem.

Line 40 Luciola lateralis is now Aquatica lateralis (reference Fu et al. 2010 Zootaxa 2530.

Line 48 lateralis.

Line 59. The subfamily is Luciolinae.

Lines 61-64. You need a reference for this life cycle material is this actually how unmunsana behaves?? And does unmunsana have a pupal cocoon? You manage to evade saying so! If you don’t know then say so and you can use this other information as simply this is what happens in other Korean species.

Lines 66-69. You say there are differences in shape then you do not tell us what those differences are (males are elongate and slender and females are wider especially across the posterior area of the elytra; the female abdomen usually protrudes beyond the elytral apices and they are of course flightless but do they have no hind wings or just vestigial wings??

Line 66 Two light emitters???? The two terminal ventral segments of the abdomen contain the light organs in the male. In the female there is only one segment which has a light organ and it is the third segment from the tip of the abdomen.

Line 68. Did you look? Are the hind wings completely missing or are they present in a shortened form? Check your references.

Line 75 why not tell us what species they are?

Lines 94, 95. Confusing. If you collected fresh adults in the field then why would you need to remove ethanol. This seems to infer that you preserved them in ethanol in the field and subsequently needed to remove that in the lab???

Line 145 Yes the flightless female will certainly inhibit movement, but are there other ways the species could move even small distances? You do not think about the larvae.

Line 152. Identification.

Line 170 and following. I like that you had a basis for comparison from 2013.

Author Response

Thank you so much for your kind and detailed comments on our manuscript. Your comments were definitely helpful for us to modify our one. We have changed many parts depending on your comments and you will meet with a newer upgraded one.

Once again we appreciate your contribution to our manuscript.

For 1. We didn't find Kato et al. (2020). Let us have time to take the exact reference. But I didn't recognize what you meant: You do not follow taxonomic references which address the nomenclature. Would you additionally add a comment in more details? Thank you in advance.

For 2. You are right. I am an ecologist, not a specialist in entomology. But I am so curious about its phenomenon. Thanks^^

For 3. You are right. Since we completed one research at the end of 2015, we have stopped continuing this research.  That is why we followed old version for this. We modified all of naming for Hotaria to Luciola.

For 4. We changed that.

For 5. Please let us have additional time to check the exact name for Luciola lateralis.

For 6. We took a research from 2013 to 2015 with preliminary research in 2012. Before starting the research and during the research period, we had screened and confirmed where there were L. unmunsana. 10 regions were the finally confirmed regions for the species. We also mentioned this briefly in our manuscript.

For 7. We checked the number twice at night. Especially, the appearance period of L. unmunsana is slightly different from the one of A.(?) lateralis. And the light colors of both are also different together. L. unmunsana has the color from light bulb (relatively orange-like color), while A. lateralis has the color from fluorescent light (slight white-like color)

Line 3: We changed it.

Line 24. We changed it to ‘as a result.

Line 27. We didn't think this line is useful for our conclusion. So we removed this line.

Lines 33-39. We changed a lot parts of these lines.

Line 40: Isn't it true? We will check one more time to change this line. But we need additional time to confirm it: Luciola lateralis is now Aquatica lateralis (reference Fu et al. 2010 Zootaxa 2530.)

Line 48 lateralis? (We didn't see this word on line 48). Please check it one more time.

Line 59. This was modified a lot.

Lines 61-64. It is our mistake on the statement of the pupal cocoon. They didn't make any type of cocoon in the pupa stage. We removed this line and modified the contents around the line.

Lines 66-69. We mentioned about sexual dimorphism of L. unmunsana. please check lines 54 to 62 in a newly upgraded one.

Line 66: Thanks a lot for your kind recommendation. Your statement should be more professional. We followed your direction(Refer to lines 58 to 60).

Line 68. Did you look? Are the hind wings completely missing or are they present in a shortened form? Check your references.

Answer to Line 68: We didn't see it exactly. But I believe the female doesn't have hind wings. But I didn't find a proper reference on this. Let us have additional time to share the best one with you.

Line 75 why not tell us what species they are?

Answer: It is not clear and furthermore there have been debating among experts. So we have been trying to follow the lately upgraded information on its number in Korea.

Lines 94, 95. Confusing. If you collected fresh adults in the field then why would you need to remove ethanol. This seems to infer that you preserved them in ethanol in the field and subsequently needed to remove that in the lab???

Answer: You are surely correct. We slightly modified this statement on lines 100 to 102.

Line 145 Yes the flightless female will certainly inhibit movement, but are there other ways the species could move even small distances? You do not think about the larvae. 

Answer: That is the reason why we tried to find the genetic perspectives on its mtDNA depending on spatial location. Even though the larvae can make a small shift in the distance, they can not cross the river and I can not imagine they can climb across big mountains.

Reviewer 3 Report

Dear authors, 

I believe your research should be improved and be published in the future as a relevant contribution if the authors decide to make a major revision. 

The main problem I observed is that it lacks a theoretical foundation. There is recent literature that should be cited in the introduction and discussed. You superficially mention the issue related to reducing the populations of fireflies without citing a single reference (see for example Vaz et al. 2021 and references therein). This issue should be further developed in the discussion, concerning which factors are associated with the reduction of fireflies in South Korea. At least preliminarily, so that, in fact, "the findings of this study can be used as baseline data for re-introducing H. unmunsana" as the authors stated in the abstract.

In the introduction, you briefly mention the taxonomic problems and delimitation of the species that have been addressed in the literature but seem to be unaware of the most recent works (e.g. Hang et al., 2019, Jusoh et al. 2021 and references therein). Because of the problematic taxonomy of L. (H). unmunsana, it is necessary to clarify in which sense H. unmunsana is employed. How have you identified the distinct populations? Otherwise, they compromise their conclusions.

Most studies recognized Hotaria as a subgenus of Luciola (See Han et al. 2019 and references therein). Luciola (Hotaria) belongs to Luciolinae subfamily. Luciola lateralis is sister-group of L. (Hotaria), a clade within Luciola according to Jusoh et al. 2021, which supports the status of Hotaria as a subgenus of Luciola. Therefore, the adoption of Hotaria as a valid genus needs justification.

It is said that "eight firefly species have been recorded in the Korean Peninsula, only three species of light-emitting fireflies are commonly seen today". Which research indicated this statement? Without references, this seems to be based on anecdotal evidence. See also Kang (2012), who listed "six species in five genera and two subfamilies are known in Korea (apud Han et al. 2019).

The results were not discussed in the light of the previous knowledge and the species delimitation problem was not addressed.

By avoiding addressing the problem of species delimitation in Luciola (Hotaria) from South Korea in the discussion, I believe you seriously compromise their conclusions regarding both the density and the relationships between the populations. Density data is presented without any context. Readers are not informed about the environments and conditions in which the males were spotted, for example, if there was light pollution. or, taking into account the biology of the species, are the larval habitats preserved?

The results were not interpreted considering the different ways in which habitats have been changed. 

Please, kindly find some comments in the file as well.

Vaz et al. https://doi.org/10.1111/icad.12481

Jusoh et al. 2021 https://doi.org/10.3390/ani11030687

Hang et al. doi:10.1163/18759866-20191420

Author Response

Dear reviewer:

Thank you so much for your detailed comments. We modified most of the region in the introduction with your recommendation. And we also upgraded the nomenclature for the species with newly upgraded references. 

And we also added the exact reference on the statement: "eight firefly species have been recorded in the Korean Peninsula, only three species of light-emitting fireflies are commonly seen today.

And finally, we have touched many portions of every part of our manuscript based on your comments. 

Once again, Thank you so much for your exact comment on the problem of our manuscript.

Round 2

Reviewer 3 Report

Dear authors
I appreciate your effort to improve the work. However, the work still needs corrections, mainly because you presented new hypotheses of correlation between soil type or environmental humidity with population density, without clearly stating a justification based on a background in the introduction, without explaining how the data was obtained in the methodology section and without making a proper analysis. 

Please, find my comments in the MS.

Author Response

Dear reviewer:

Thank you for all comments and kind guidance. We have checked all of your comments, and modified all kind of things which, you considered, should be upgraded. Now we are happy that we could make much more upgraded manuscript because of your comments. Once again, we appreciate your contribution and we also hope this manuscript would be getting you satisfied a lot. We are also waiting for your positive response. 

Sincerely yours,

Gab-Sue Jang